# Cochlear Ribbon Synapses in Aged Gerbils

**DOI:** 10.3390/ijms25052738

**Published:** 2024-02-27

**Authors:** Sonny Bovee, Georg M. Klump, Sonja J. Pyott, Charlotte Sielaff, Christine Köppl

**Affiliations:** 1Department of Neuroscience, School of Medicine and Health Science, Carl von Ossietzky Universität Oldenburg, 26129 Oldenburg, Germany; sonny.bovee@uol.de (S.B.); georg.klump@uni-oldenburg.de (G.M.K.); charlotte.sielaff@item.fraunhofer.de (C.S.); 2Cluster of Excellence “Hearing4all”, Carl von Ossietzky Universität Oldenburg, 26129 Oldenburg, Germany; 3Research Centre Neurosensory Science, Carl von Ossietzky Universität Oldenburg, 26129 Oldenburg, Germany; 4Department of Otorhinolaryngology/Head and Neck Surgery, University Medical Center Groningen, University of Groningen, P.O. Box 30.001, 9700 RB Groningen, The Netherlands; s.pyott@umcg.nl; 5Fraunhofer Institute for Toxicology and Experimental Medicine (ITEM), 30625 Hannover, Germany

**Keywords:** hearing, auditory, cochlea, synaptopathy, age-related hearing loss, auditory nerve, inner hair cell, gerbil

## Abstract

In mammalian hearing, type-I afferent auditory nerve fibers comprise the basis of the afferent auditory pathway. They are connected to inner hair cells of the cochlea via specialized ribbon synapses. Auditory nerve fibers of different physiological types differ subtly in their synaptic location and morphology. Low-spontaneous-rate auditory nerve fibers typically connect on the modiolar side of the inner hair cell, while high-spontaneous-rate fibers are typically found on the pillar side. In aging and noise-damaged ears, this fine-tuned balance between auditory nerve fiber populations can be disrupted and the functional consequences are currently unclear. Here, using immunofluorescent labeling of presynaptic ribbons and postsynaptic glutamate receptor patches, we investigated changes in synaptic morphology at three different tonotopic locations along the cochlea of aging gerbils compared to those of young adults. Quiet-aged gerbils showed about 20% loss of afferent ribbon synapses. While the loss was random at apical, low-frequency cochlear locations, at the basal, high-frequency location it almost exclusively affected the modiolar-located synapses. The subtle differences in volumes of pre- and postsynaptic elements located on the inner hair cell’s modiolar versus pillar side were unaffected by age. This is consistent with known physiology and suggests a predominant, age-related loss in the low-spontaneous-rate auditory nerve population in the cochlear base, but not the apex.

## 1. Introduction

Hearing begins in the cochlea of the inner ear where hair cells transduce the physical sound stimulus. The afferent neurons synaptically connected to them transmit this auditory information to the central nervous system. In the mammalian cochlea, type-I afferent auditory nerve fibers comprise the vast majority of afferents (~95%; [1]). They are connected to inner hair cells (IHCs) via specialized ribbon synapses comprising a presynaptic ribbon and postsynaptic glutamate receptor patch [2]. About 15–20 auditory nerve fibers, with different physiological and molecular identities, connect to a given IHC, and their synapses are also subtly different (reviewed by [3,4,5]). Classic cat studies showed that low-spontaneous-rate (SR) fibers typically connect on the modiolar side of the IHC, while high-SR fibers are typically found on the pillar side, with synapses on the modiolar side having larger ribbons and smaller axons [6,7,8]. Since then, salient differences in the Ca^++^ dynamics of modiolar versus pillar ribbon synapses were observed in mouse IHCs in vitro, that help to explain the different in-vivo physiology of high- and low-SR auditory nerve fibers [5]. Furthermore, gradients in the morphological properties of ribbon synapses and associated afferent fibers around the IHC were confirmed and extended for several mammalian species and are now generally believed to be typical correlates of auditory nerve physiological types in the mammalian cochlea. Their specific role for shaping afferent fiber physiology, however, remains unclear. Similar to the cat, a difference in axon diameters was also found for high- versus low-SR auditory nerve fibers in guinea pig [9,10]. A gradient in presynaptic volume was confirmed for all mammals investigated: in different mouse strains [11,12], in guinea pig [13], in rat [14], in gerbil [15] and in mole rat [16]. More recent quantitative immunohistochemical studies found additional gradients in the volumes of postsynaptic glutamate receptor (GluR) patches that, however, proved to be more species-specific. In gerbils, modiolar-located synapses with larger presynaptic ribbons also showed larger postsynaptic GluR patches [15], while in mice, depending on the strain, the postsynaptic volumes showed either an opposing or concurrent gradient compared to the presynaptic ribbon volume gradient [11,12]. The continued maintenance of this heterogeneity in ribbon synaptic morphology, and thus presumably afferent fiber physiology, may depend on the presence of olivocochlear efferent innervation. In mice, de-efferentation by sectioning the olivocochlear (OC) outgoing fiber bundle was shown to eliminate all spatial gradients in synaptic elements [17]. In the cat, similar de-efferentation caused a decrease in average spontaneous firing rates and changed the distribution across SR classes [18].

Loss of auditory nerve synapses on IHCs, often referred to as cochlear synaptopathy, occurs gradually with advancing age and, in addition, is triggered by noise exposure. Cochlear synaptopathy is believed to underlie early-stage hearing deficits that humans commonly report but that are not detected by standard clinical audiometry [19]. To identify the functional consequences of cochlear synaptopathy, it is important to clarify whether such loss of ribbon synapses happens randomly or whether the physiological subtypes of type-I fibers are differentially affected. Currently, there is intriguing evidence for preferential, disproportionate loss in the low-SR population [13,20,21,22,23,24,25]. However, perceptual deficits that may arise from this have been difficult to identify in humans and remain controversial [26,27,28,29]. Animal studies that aim to directly correlate cochlear synaptopathy and physiology or perception remain rare and have yielded mixed results [13,30]. Furthermore, intrinsic repair processes may reinstate functional synapses, but the extent and time course of such repair remain unclear. It has so far only been demonstrated after noise exposure (not with aging) and may depend on the species investigated as well as the details of the trauma triggering the synaptopathy [23,31,32,33]. Finally, compensatory mechanisms at surviving synapses have also been observed [34].

In this study, we explored cochlear synaptopathy in aged Mongolian gerbils (*Meriones unguiculatus*), a well-established animal model of age-related cochlear degeneration and hearing loss (reviewed by [35,36]). The main advantages of the gerbil are its relatively short lifespan, a hearing range that includes lower frequencies that overlap with that of humans, and its amenability to behavioral training for psychoacoustic tests [37]. Moreover, our gerbils live their entire life in a controlled, quiet environment, which minimizes potentially confounding effects of noise and ototoxic substances. Gerbils are already known to develop synaptopathy, or a reduction in functional synapse numbers on IHCs, with advancing age [38,39]. In addition, comparisons of single-unit data from young-adult and quiet-aged gerbils suggested an age-related, disproportionate loss of low-SR fibers in auditory nerve fibers tuned to high frequencies [20,25]. The aim of the present study was to explore whether these losses affect the spatial distribution of synapses and/or the morphological gradients in the volumes of synaptic elements along the modiolar–pillar axis of their IHCs. In other words, do the suggested correlations of synaptic morphology and physiological identity hold across the lifespan? If so, this would further support a functional role of these gradients in shaping the individual synapse’s transmission properties.

## 2. Results

Three cochlear locations were chosen for detailed analysis, corresponding to 1, 2 and 16 kHz. Since the previous analysis in young adults [15] had not revealed any pronounced tonotopic variation in synapse numbers or volumes, we decided to focus on two low-frequency locations in a range that is important for speech processing in humans, and one high-frequency location. In the present study, each cochlear location is represented by six to eight individual animals, with a median number of nine IHCs evaluated from each (for details see Appendix A). In total, we report results from 376 IHCs. Figure 1 and Figure 2 show representative examples from a young-adult (Figure 1) and a quiet-aged (Figure 2) gerbil, both at a location corresponding to 16 kHz.

### 2.1. Age-Related Loss of Synapses

Averaged across all three cochlear locations, young-adult gerbils had a mean of 21.6 functional synapses (paired pre- and postsynaptic elements) per IHC, while quiet-aged gerbils showed a mean of 17.2, a reduction of 20.4%. A mixed-model ANOVA showed a significant effect of the fixed factors age group and frequency location on synapse number per IHC ([age group] F(1,38) = 36.705, *p* < 0.001, [frequency] F(2,38) = 6.878, *p* = 0.003). Pairwise post-hoc tests, however, could not confirm tonotopic differences when combining data from young-adult and quiet-aged gerbils in that analysis. Only the 1 kHz and 2 kHz cochlear locations differed significantly from each other with respect to synapse number. Comparing the age groups, however, synapse numbers were significantly reduced at the 2 and 16 kHz position in the quiet-aged individuals compared to young-adult individuals (Figure 3). No interaction between age group and frequency was found, indicating similar variation among the three locations in both age groups, and similar losses with aging.

### 2.2. Age-Related Change in the Spatial Distribution of Synapses on the IHC

The 20+ synapses that contact a typical IHC in young adults originate from auditory afferent neurons with different morphology (and probably also different physiology) that tend to segregate along the pillar–modiolar axis of the IHC [15]. We therefore explored whether the distribution of synapses around the IHC changed with age. Synapses were grouped into “modiolar” or “pillar”, depending on where their center of gravity fell along the individual IHC’s pillar–modiolar axis (examples in Figure 1C–E and Figure 2C–E). Figure 4 shows the distribution of all evaluated synapses along this IHC axis. Separate counts of modiolar and pillar synapses were then obtained for each IHC (Figure 5). A mixed-model ANOVA with synapse number as the dependent variable and age group, frequency, and side on the IHC (modiolar versus pillar) as fixed factors, showed a significant effect of age group ([age group] F(1,76) = 10.064, *p* < 0.001), a significant two-way interaction of frequency and side ([frequency*side] F(2,76) = 3.380, *p* = 0.039), and a significant three-way interaction of age group, frequency and side ([age group*frequency*side] F(2,76) = 5.410, *p* = 0.006). This finding suggested different synapse distributions on IHCs of the different cochlear locations, and differential changes in these distributions with age. To further explore these effects, we conducted separate ANOVAs at the three different frequency locations. At the 16 kHz location, a significant effect of age ([age group] F(1,26) = 10.651, *p* = 0.003), side on the IHC ([side] F(1,26) = 5.876, *p* = 0.023), and a significant interaction between age and side on the IHC ([age group*side] F(1,26) = 22.511, *p* < 0.001) were found. In young-adult gerbils, the number of synapses was higher on the modiolar side than on the pillar side (mean of 13.7 modiolar and 8.3 pillar, total of 22), whereas in quiet-aged gerbils, this distribution reversed and the pillar side had higher synapse numbers than the modiolar side (mean 7.7 modiolar and 9.4 pillar, total of 17.1). Thus, at the 16 kHz position, quiet-aged gerbils had, on average, lost 4.9 synapses per IHC, and this entire loss can be explained by losses on the modiolar side of the IHC. This strongly suggests that quiet-aged gerbils lost synapses disproportionally more on the modiolar side at the 16 kHz position (Figure 5C). Post-hoc testing confirmed that compared to young-adult gerbils, at the 16 kHz location, quiet-aged gerbils had significantly fewer synapses on the modiolar side of the IHC (Figure 5C). At the 1 and 2 kHz positions, neither an effect of age, nor side on the IHC, nor an interaction between age and side on the IHC was found. Thus, despite similar overall synapse losses at all cochlear locations, a differential loss that changed the distribution of synapses contacting the modiolar or pillar side of the IHC was only found at the base of the cochlea.

### 2.3. Pre- and Postsynaptic Volume Showed Only Minor Changes with Aging

In young-adult gerbils, presynaptic ribbon volumes and the volumes of postsynaptic GluR patches vary concurrently with position around the IHC, such that synapses located on its modiolar side tend to show larger volumes for both synaptic partners than synapses located on the pillar side [15]. At all three cochlear locations, in both young-adult (Figure 6A–C) and quiet-aged gerbils (Figure 6D–F), volumes of the pre- and postsynaptic elements were also correlated in the present sample. To explore whether the distribution of synapse volumes around the IHC was affected by age, we conducted a mixed-model ANOVA with normalized volume as the dependent variable, and age, frequency position, and side on the IHC as fixed factors. Separate analyses were conducted for the pre- and postsynaptic elements. For the presynaptic elements, the ANOVA showed a significant effect of the position on the IHC ([pillar-modiolar] F(1,76) = 44.908, *p* < 0.001), as well as an interaction between the side of the IHC and the frequency location ([frequency*pillar-modiolar] F(2,76) = 7.384, *p* = 0.001). This interaction reflects that the volume difference between the presynaptic ribbons at the modiolar and pillar side appeared to increase with increasing frequency (Figure 7A–C); it was statistically significant only for the 16 kHz location (Figure 7C). There was no significant difference between age groups ([age group] F(1,76) = 2.388, *p* = 0.126). This confirms a differential distribution of ribbon volumes around the IHC and suggests differences between cochlear locations, but no changes with aging. For the postsynaptic elements, the ANOVA also showed a significant effect of the position on the IHC ([pillar-modiolar] F(1,76) = 42.765, *p* < 0.001), as well as an interaction between the side of the IHC and the frequency location ([frequency*pillar-modiolar] F(2,76) = 8.882, *p* < 0.001)). Similar to the presynaptic elements, this interaction reflects that the volume difference between the postsynaptic GluR patches at the modiolar and pillar side appeared to increase with increasing frequency (Figure 7D–F). For the postsynaptic (but not the presynaptic) elements, the ANOVA additionally revealed an interaction between side and age group ([side*age group] F(2,76) = 11.524, *p* = 0.001), suggesting an age effect. This interaction reflects that the volume difference between postsynaptic GluR patches on the pillar and modiolar side was larger for quiet-aged gerbils than for young-adult gerbils. In summary, the effect of age on synapse volume was minor and did not suggest deterioration, but rather a sharpening of the typical volume gradients in synaptic elements around the IHC.

We were unable to explore changes in absolute synaptic volumes with age. Absolute volume measurements are not possible from immunofluorescent material. Thus, relative comparisons between specimens require strictly comparable specimen processing, image acquisition, and image analysis [40,41]. Sufficient material from quiet-aged gerbils in particular was only available to us over prolonged periods of time. Cochleae were therefore processed individually and the different steps were carried out by different researchers over time. The influence of individual researchers on absolute synaptic volumes was clearly evident, and we therefore abstained from further analysis.

## 3. Discussion

As briefly reviewed in the Introduction, type-I afferent auditory nerve fibers are typically divided into physiological types based on their spontaneous firing rate and response properties to auditory stimuli. Low-SR fibers connect preferentially on the modiolar side of the IHC, high-SR fibers on the pillar side, and a number of gradients in synaptic morphology along the modiolar–pillar IHC axis correlate with this physiological heterogeneity (reviewed by [3,4,5]). In this study, we investigated, at different tonotopic locations along the organ of Corti, whether age-related synapse loss (synaptopathy) affects the spatial distribution of synapses or the morphological gradients in the volumes of synaptic elements along the modiolar–pillar axis of their IHC. Synapse numbers were significantly lower in quiet-aged individuals, and at apical cochlear locations (equivalent to 1 and 2 kHz best frequency) this loss occurred equally on both the modiolar and pillar sides of the IHC. However, at the basal cochlear location (equivalent to 16 kHz), the loss was clearly selective and occurred predominantly on the modiolar side of IHC. However, the typical differences in the volumes of pre- and postsynaptic elements on the modiolar and pillar sides of the IHC remained unchanged with age.

### 3.1. Age-Related Loss of Synapse Numbers

To probe for the occurrence of synaptopathy with age in gerbils, we compared the number of functional synapses per IHC between young-adult and quiet-aged gerbils. The mean number of synapses per IHC was reduced by 20.4% in quiet-aged individuals, which agrees very well with previous studies in gerbil [38,39]. Similar reductions with aging were also reported for rat [42], whereas humans [43] and CBA/CaJ mice [44] showed more severe losses. We could not confirm a differential age-related loss along the tonotopic gradient as previously suggested for gerbils [38,39]. However, those previous reports did not agree on the region of maximal loss, and there is as yet no consistent pattern across all available studies in different species [38,39,42,43]. Discrepancies across studies may arise from differences in the relative age that is studied along the species’ lifespan, and in the acoustic environment during aging [39,44].

### 3.2. Differential Loss of IHC Afferent Subtypes with Age

We next investigated whether synaptopathy in aging gerbils differentially affects the numbers of synapses located on the modiolar or pillar side of the IHC. Compared to young adults, we found that at the 16 kHz position near the base of the cochlea, the distribution of synapses around the IHC had significantly changed in quiet-aged gerbils, such that fewer synapses were observed on the modiolar side. Lower-frequency locations near the apex did not show such a change. These results strongly suggest a disproportionate loss of synapses associated with low-SR fibers near the base of the cochlea. This is consistent with the previous observation of changed SR distribution in populations of auditory nerve fibers recorded in young-adult and quiet-aged gerbils [20,25]. Together, these data now show a clear pattern of an age-related, disproportionate loss of suspected low-SR fibers, specifically in auditory nerve fibers tuned to high frequencies (>4 kHz).

In mice, the data are less clear and partly contradictory. In aging CBA/CaJ mice, presumed low-SR auditory nerve fibers, which were identified by their gene expression profile, diminished disproportionately [22]. However, also in CBA/CaJ mice, noise exposure resulted in synaptopathy in the basal half of the cochlea that affected both low- and high-SR fibers non-selectively [30]. Ultrastructurally, such mice displayed mild to severe synaptic abnormalities, mainly on the modiolar side of the IHC [45], suggesting early stages of selective degeneration. In a large-scale study of aging mice of four different strains (not including CBA/CaJ), loss of ribbon synapses tended to occur equally on the pillar and the modiolar side of the IHC; but one strain showed evidence for a preferential loss on the pillar side [34], i.e., the reverse of our observations in gerbil. Finally, in aging C57Bl/6J mice, a preferential loss of synapses on the surface of the IHC facing the cochlear base (the high-frequency surface) was reported [46], a different pattern of differential loss that does not obviously relate to physiology.

In guinea pig, compared to control animals, a loss of low-SR auditory nerve fibers was observed in neurophysiological population data, at least in the high-frequency region (>4 kHz) [13]. This was as a consequence of noise exposure and agrees with our findings in aged gerbils. There are also indirect indications that loss of activity, specifically of low-SR fibers, occurs in older humans [24].

In summary, the evidence for a disproportional loss of modiolar-located ribbon synapse associated with low-SR auditory nerve fibers is substantial, both for aging IHC and after noise exposure, but it is not unequivocal. It remains to be clarified whether the contradictory findings in different mouse studies may reflect subtle differences in experimental conditions or techniques. We caution that indirect metrics of synaptopathy can be confounded by changes in the discharge behavior of surviving auditory nerve fibers [30], in particular in aging individuals with mixed cochlear pathologies [39].

### 3.3. Volume Gradients of Synaptic Elements Remain with Aging

In young-adult gerbils, volumes of pre- and postsynaptic elements showed a positive correlation, with larger synaptic elements located on the modiolar side of the IHC, extending our previous findings in a smaller sample of young-adult gerbils [15]. We add here that this size difference became more prominent with increasing equivalent frequency, toward the base of the cochlea. Compared to young adults, the results for quiet-aged gerbils were very similar, with only a slightly enhanced difference in volume between pillar and modiolar postsynaptic elements. Thus, the subtle gradients in synaptic morphology around the IHC are robust across the lifespan of gerbils. This may differ from synaptopathy induced by noise exposure, where the characteristic volume gradients of synaptic ribbons around the IHC were observed to be transiently disrupted in guinea pigs [13,33]. In mice, immediately after noise exposure, however, the volume difference between modiolar- and pillar-located ribbons was actually enhanced [45], similar to our finding in quiet-aged gerbils.

It is important to note that our analysis always used normalized volumes of synaptic elements. Thus, the finding that the difference in volumes between modiolar- and pillar-located synaptic elements was robust across ages does not exclude the possibility that changes in absolute volumes occurred. In IHC of aged mice, the surviving ribbons showed an increase in absolute volume and the kinetics of Ca^2+^-dependent exocytosis remained unchanged, indicating perhaps a form of functional compensation for the overall loss in ribbon number [34]. Consistent with that, compared to young adults, the ultrastructure of surviving ribbon synapses in aged mice showed enlarged postsynaptic afferent terminals with enlarged mitochondria and enlarged postsynaptic densities, as well as enlarged presynaptic bodies [46]. Similarly, after noise-induced synaptopathy in mice, enlarged ribbons, as well as a drastic increase in synapses with multiple ribbons, were observed in surviving synaptic contacts [45]. Interestingly, even in young-adult mice, the mean size of ribbons in an IHC was found to be negatively correlated with its total number of ribbons, suggesting a fixed amount of available ribbon material per IHC [47]. This may not apply to outer hair cells in which an increase in number of ribbons, as well as ribbon volume, was found after noise exposure in mice [48]. A further intriguing observation is that olivocochlear efferents may be involved in the regulation of afferent synaptic morphology. Transient changes in the number and size of efferent contacts on type-I auditory nerve fibers were observed within a short interval after noise exposures that triggered afferent synaptopathy [45]. Changes in efferent innervation also occurred on IHCs of aged mice [49].

In summary, the subtle differences in the volumes of synaptic elements located on the modiolar versus the pillar side of the IHC are quite robust with advancing age and, if at all, are probably also only transiently disrupted after noise exposure. This is consistent with the persistence of the basic physiological types of auditory nerve fibers into old age and after noise exposure. Despite this, the evidence of changes in absolute volumes which may occur to compensate for the loss in number is intriguing. The mechanisms driving these changes in surviving synapses, as well as their functional consequences remain to be clarified.

## 4. Materials and Methods

### 4.1. Animals

A total of 22 Mongolian gerbils (*Meriones unguiculatus*) were used for this study. A total of 9 animals (6 females, 3 males), aged between 80 and 319 days (median 205 days or 6.7 months), were defined as “young-adult”, and a further 13 animals (4 females, 9 males), aged between 36 and 41.7 months (median 39.9 months), were defined as “quiet-aged” (see also Appendix A). All animals were born in the animal facility of the University of Oldenburg and lived their entire lives in a controlled, quiet environment. All protocols and procedures were approved by the authorities of Lower Saxony, Germany, permits AZ 33.19-42502-04-15/1828 and AZ 33.19-42502-04-15/1990.

### 4.2. Cochlear Processing

For full details of the procedures see Steenken et al. [39]. Animals were euthanized by an overdose of sodium pentobarbital. Tissue fixation with 4% paraformaldehyde in phosphate-buffered saline (PBS) was achieved either by transcardial perfusion or cochlear immersion after decapitation and rapid extraction of bullae. After a total time of typically 2 days in fixative, the isolated bullae were transferred to 0.5 M EDTA in PBS for decalcification, taking typically 2 days with constant agitation at 8 °C.

The cochlear tissue was then permeabilized with 1% Triton X-100 in PBS, and unspecific protein binding sites were blocked with 3% bovine serum albumin (BSA) with 0.2% Triton in PBS for 1 h at room temperature. An anti-CtBP2 (C-terminal binding protein 2) antibody (IgG1 monoclonal mouse; BD Biosciences, Heidelberg, Germany; cat. no. 612044; RRID: AB_399431) at a concentration of 1:400 was used to label presynaptic ribbons. An anti-GluR2 antibody (IgG2a monoclonal mouse; Merck-Millipore, Darmstadt, Germany; cat. no. MAB 397; RRID: AB_2113875) at a concentration of 1:200 labeled postsynaptic receptors. An anti-Myosin VIIa antibody (IgG polyclonal rabbit; Proteus Biosciences, Waltham, MA, USA; cat. no. 25-6790; RRID: AB_10015251) at a concentration of 1:200 labeled hair cells. All 3 primary antibodies were diluted in the blocking solution, and the cochleae were incubated in this mix at 37 °C overnight. Three secondary antibodies, matching the hosts of the primary antibodies—goat anti-mouse (IgG1)-AF 488 (Life Technologies-Molecular Probes, Eugene, OR, USA; cat. no. A21121; RRID: AB_141514) at a concentration of 1:500, goat anti-mouse (IgG2a)-AF568 (Invitrogen, Carlsbad, CA, USA; cat. no. A-21134; RRID: AB_10393343) at a concentration of 1:200, and donkey anti-rabbit-AF647 (polyclonal secondary antibody; Life technologies-Molecular Probes, Eugene, OR, USA; cat. no. A-31573; RRID: AB_162544) at a concentration of 1:500—were used to bind specifically to the 3 primary antibodies. Cochleae were incubated with the secondary antibodies diluted in blocking solution at 37 °C for 2 h.

After the conclusion of immunolabeling, the majority of cochleae from quiet-aged gerbils (12 of 14 cochleae from 13 gerbils) was briefly treated with the autofluorescence quencher TrueBlack (TrueBlack Lipofuscin Autofluorescence Quencher, 20× in DMF, Biotum, Hayward, CA, USA) (5% in 70% ethanol, for one minute).

Finally, cochleae were microdissected into typically 6–8 pieces of organ of Corti that were flat-mounted in Vectashield Antifade Mounting Medium (Vector Laboratories, Burlingame, CA, USA) on a microscope slide, coverslipped, and sealed with nail varnish.

### 4.3. Image Acquisition and Processing

For full details, see again Steenken et al. [39]. Every cochlear segment was examined and documented using a Nikon epi-florescence microscope system (Nikon 90i with NIS Elements software, Version 4.30) and a 4× objective. The length of the entire cochlea was measured and three different cochlear locations, at 2.51, 3.8 and 8.13 mm from the apex, corresponding to 1, 2, and 16 kHz, respectively [50], were selected for the synapse analysis. These locations were examined with a confocal microscope (Leica Microsystem CMS GmbH, Wetzlar, Germany; Leica TCS SP8 system) using an oil-immersion objective (40x, numerical aperture 1.3 or 63×, numerical aperture 1.4). The 488 nm (OPSL), 522 nm (OPSL) and 638 nm (Diode) lasers were used to excite the three antibody-labeled fluorescent channels. Image stacks were acquired at 0.07–0.10 µm/pixel resolution and using 0.3 µm or (one cochlea only) 0.5 µm steps in the z-dimension, with sequential scanning of the three channels for each z-layer and a hybrid detector in photon-counting mode. Detector bandwidths for each channel were adjusted to exclude crosstalk.

All confocal stacks were subjected to deconvolution (Hyugens Essentials, Version 15.10, SVI, Hilversum, The Netherlands) with default settings (maximum iterations: 80; Signal to noise ratio: 10; Quality threshold: 0.01) using a theoretical point-spread function.

### 4.4. Image Analysis

A custom-written image analysis software (MATLAB 2015a, The MathWorks, Inc., Natick, MA, USA) was used to determine the numbers and volumes of synaptic elements. This procedure was developed and validated by Zhang et al. [15]. Several trained observers contributed to the full sample, and one of the authors (CK) who had co-developed the procedure performed a final check for consistency.

In a first step, using the sum of all labeled channels, the borders of individual hair cells were interactively delineated, and their anatomical axes individually defined in the cochlear apex-to-base, modiolar–pillar and hair-cell top-to-bottom dimensions. Next, the synaptic elements were detected by intensity thresholding, separately for the presynaptic CtBP2-channel and the postsynaptic GluR2 channel. This could optionally be refined in several ways and was guided by 3D displays of hair cells and binarized synaptic elements. Typically, a minimum voxel count of 5 was defined (corresponding to a volume of about 0.003 µm^3^), and several elements that appeared to be two or more merged ribbons or GluR patches were interactively split.

Colocalization of pre- and postsynaptic immunolabel was used as the definition of a functional synapse. The colocalization algorithm was distance-based, in 3D, and required that colocalized elements overlapped or at least touched at their nearest points, that is, there was a minimal distance of zero between the nearest points of colocalized elements.

The absolute synaptic volumes showed variation between observers, reflecting individual preferences for more liberal or conservative definition of the initial intensity threshold for detection of synaptic elements. Normalization to the median volume within a particular confocal stack (after Liberman et al. [11]) eliminated these differences. Pre- and postsynaptic elements were separately normalized to their respective medians, and only colocalized elements contributed to the median.

### 4.5. Statistics

All statistical procedures were carried out with the use of IBM (Armonk, NY, USA) SPSS Statistics version 29. With the kind of data collected in the present study, it is important to be aware that individual synapses do not represent independently sampled cases. If entered as such into statistical procedures, these are pseudoreplications that artificially inflate test power. Furthermore, sampling from several cochlear locations of a given cochlea is a repeated-measures design. To account for these limitations, we applied linear mixed-model analyses of variance (ANOVA) with dependent variables and factors and post-hoc tests as described in the results. For the analysis of synapse numbers, mean values, obtained from between 4 and 11 IHCs for each frequency location in the ear of a given animal (Appendix A) entered the statistics. In case of synapse volumes, we used median values for each frequency location in the ear of a given animal. The latter choice was made because synaptic volumes determined from immunofluorescent material typically show a highly skewed frequency distribution, with a tail of few large and very large values (e.g., [11,51,52]). The normalization we applied (see 4.4 above) does not affect this skewed distribution. For such cases, the median is a more appropriate measure of central tendency than the mean.

The data for young-adult gerbils previously published by Zhang et al. [15] were incorporated here and provided about half of the young-adult sample (75 of 164 IHCs). As a double-check, all statistics reported here were also carried out without these previously collected data. This analysis yielded the same significant results and conclusions, albeit with slightly different *p*-values and reduced variance. From the sample of quiet-aged gerbils, synapse counts from 17 of the 24 cochlear locations analyzed here were published previously [39]. However, since the present study aimed to add data on the volumes of synaptic elements in aged IHCs, the analysis was carried out de novo on the existing image stacks and included more hair cells from each specimen.

## Figures and Tables

**Figure 1 ijms-25-02738-f001:**
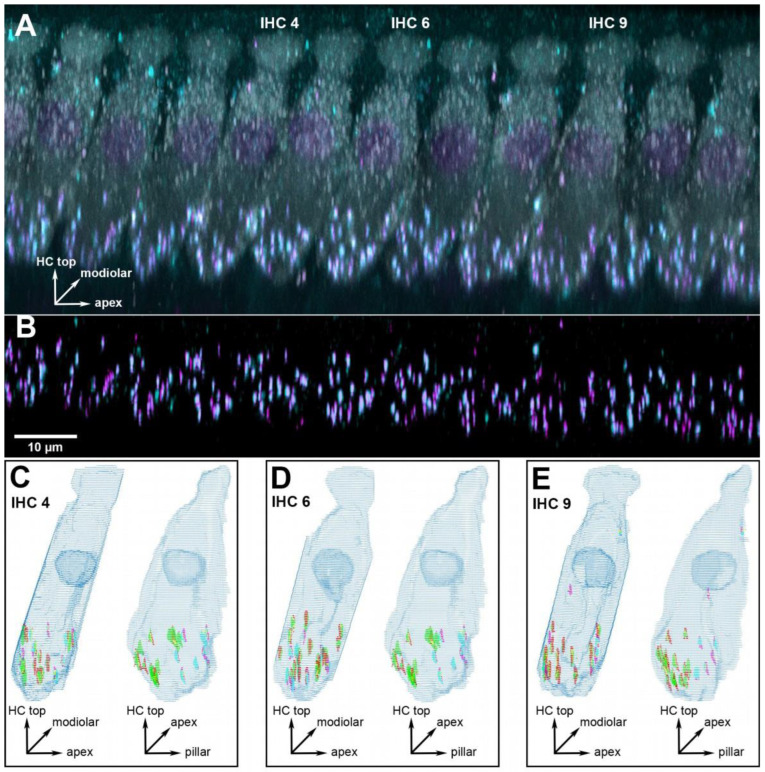
Example of a cochlear location equivalent to 16 kHz from a young-adult gerbil aged 5.1 months. (**A**) 3D projection of the raw confocal image stack (anatomical orientation indicated by the XYZ vector inset). Note that the projection was rotated 107° around the cochlear apex-to-base axis to obtain this view onto the row of inner hair cells (IHCs), i.e., the original scan direction was from bottom to top along the hair cells’ long axis. All eleven IHCs within this frame of view were evaluated, three specific ones are indicated and illustrated further in panels (**C**–**E**). Hair-cell label (anti-MyoVIIa) is shown in gray, presynaptic label (anti-CtBP2) in magenta, and postsynaptic label (anti-GluR2) in cyan. (**B**) The same view, after deconvolution, but now cropped to the lower synaptic area of the IHCs, and only showing the two synaptic labels. The scale bar applies to panels (**A**,**B**). (**C**) Grid reconstruction of IHC 4, shown from two different viewing angles (anatomical orientation indicated by XYZ vector insets): as in panel (**A**) (left image) and rotated by 90° (right image). The colocalized synaptic elements that were detected and classified for this IHC are color-coded: presynaptic modiolar is shown in red, postsynaptic modiolar in green, presynaptic pillar in magenta, and postsynaptic pillar in cyan. (**D**,**E**) The same views for IHC 6 and 9.

**Figure 2 ijms-25-02738-f002:**
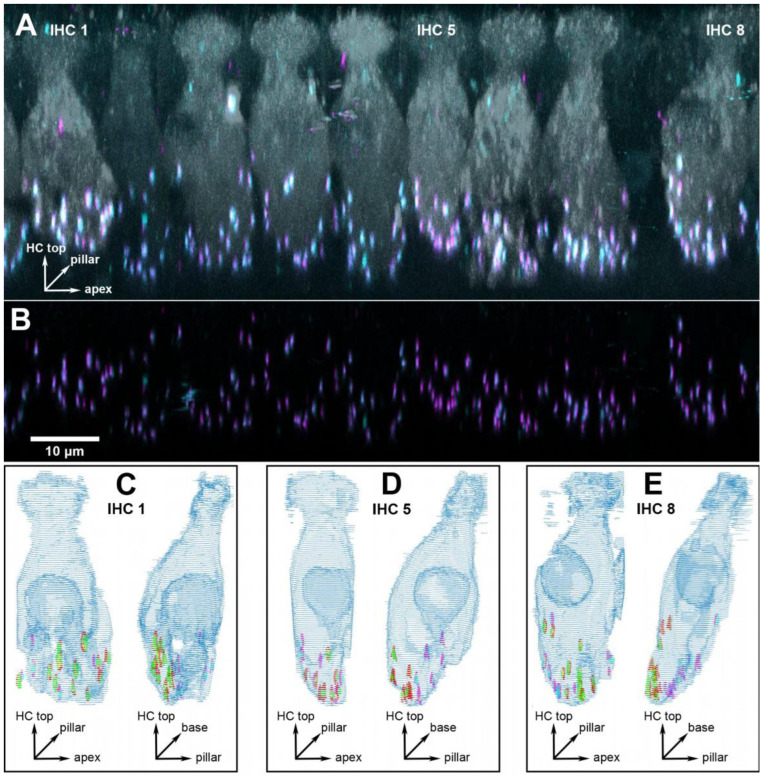
Example of a cochlear location equivalent to 16 kHz from a quiet-aged gerbil aged 36 months. (**A**) 3D projection of the raw confocal image stack (anatomical orientation indicated by the XYZ vector inset). Note that the projection was rotated 73° around the cochlear apex-to-base axis to obtain this view onto the row of inner hair cells (IHCs), i.e., the original scan direction was from bottom to top along the hair cells’ long axis. Eight IHCs within this frame of view were evaluated, three specific ones are indicated and illustrated further in panels (**C**–**E**). Hair-cell label (anti-MyoVIIa) is shown in gray, presynaptic label (anti-CtBP2) in magenta, and postsynaptic label (anti-GluR2) in cyan. (**B**) The same view, after deconvolution, but now cropped to the lower synaptic area of the IHCs, and only showing the two synaptic labels. The scale bar applies to panels (**A**,**B**). (**C**) Grid reconstruction of IHC 1, shown from two different viewing angles (anatomical orientation indicated by XYZ vector insets): as in panel (**A**) (left image) and rotated by 90° (right image). The colocalized synaptic elements that were detected and classified for this IHC are color-coded: presynaptic modiolar is shown in red, postsynaptic modiolar in green, presynaptic pillar in magenta, and postsynaptic pillar in cyan. (**D**,**E**) The same views for IHC 5 and 8.

**Figure 3 ijms-25-02738-f003:**
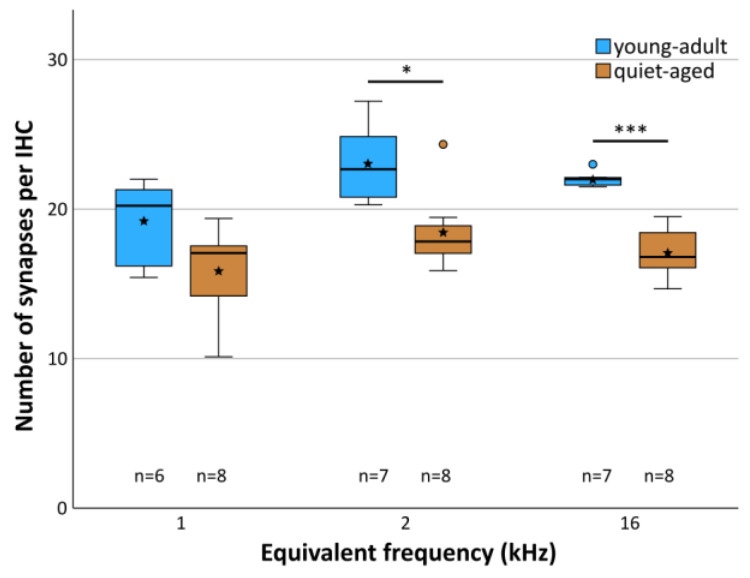
Mean number of synapses per IHC at each frequency location, separately for young adults (blue shade) and quiet-aged gerbils (brown shade). Boxes indicate the median ± interquartile range, whiskers 1.5 times the interquartile range, outliers beyond those are shown as circles (>1.5, ≤3 times interquartile range). Stars indicate means. Black asterisks above the boxes indicate statistically significant differences between data for young-adult and quiet-aged gerbils, as indicated by line brackets (* *p* ≤ 0.05, *** *p* ≤ 0.001).

**Figure 4 ijms-25-02738-f004:**
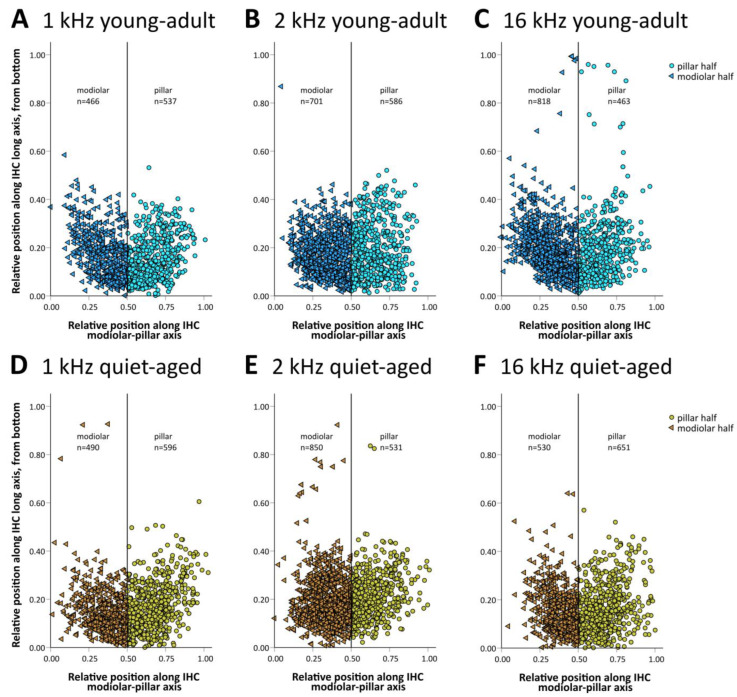
Two-dimensional distribution of synapses with respect to the IHC’s top–bottom and modiolar–pillar axes. Position was normalized to the individual IHC’s extent in both dimensions. As a visual aid, thin vertical lines separate the IHC’s modiolar and pillar half, and individual synapses located modiolar or pillar are shown with different symbols and different color shades. (**A**–**C**) Data for young-adult gerbils (in blue shades), separated according to cochlear location, from apical (equivalent to 1 kHz, (**A**)) to basal (equivalent to 16 kHz, (**C**)), as indicated. (**D**–**F**) Identical layout as in (**A**–**C**) but showing data from quiet-aged gerbils (in brown shades).

**Figure 5 ijms-25-02738-f005:**
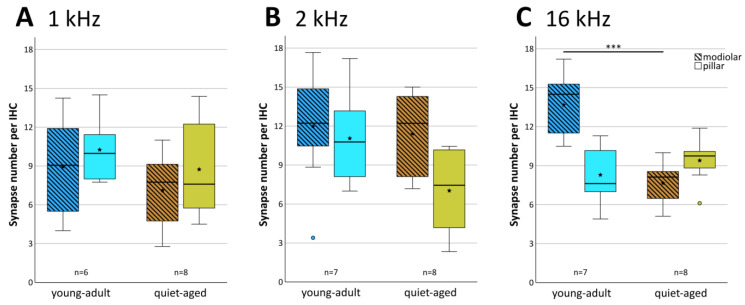
Mean number of synapses located in the IHCs’ modiolar and pillar halves. The three panels show boxplots for the three cochlear locations, equivalent to 1 kHz (**A**), 2 kHz (**B**) and 16 kHz (**C**). Each boxplot contrasts synapse numbers in the modiolar (hatched boxes) versus pillar half (clear boxes), separately for young-adult (blue shades) and quiet-aged gerbils (brown shades). Boxes indicate the median ± interquartile range, whiskers 1.5 times the interquartile range, outliers beyond those are shown as circles (>1.5, ≤3 times interquartile range). Stars indicate means. Black asterisks above the boxes in (**C**) indicate statistically significant differences between the modiolar data for young-adult and quiet-aged gerbils, as indicated by line brackets (*** *p* ≤ 0.001).

**Figure 6 ijms-25-02738-f006:**
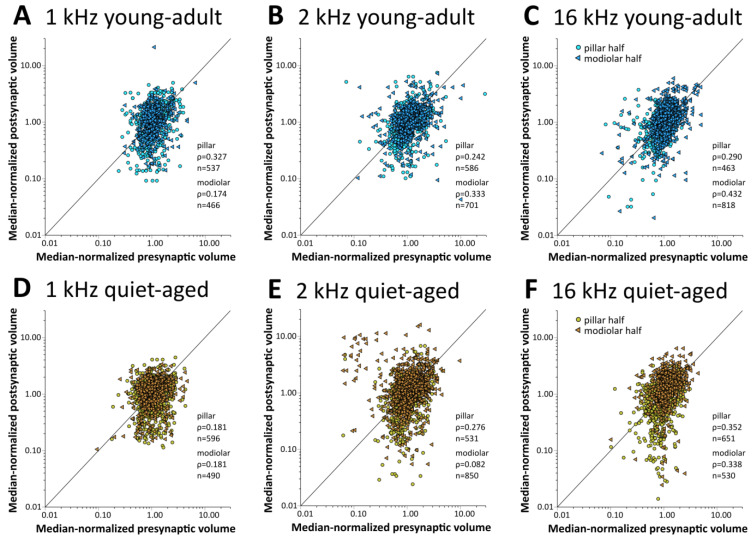
The volumes of pre- and postsynaptic elements were consistently positively correlated. Scatterplots of the normalized volume of presynaptic ribbons versus normalized volume of their postsynaptic, colocalized GluR2 receptor-patch partner. The top row of graphs (**A**–**C**) shows data for young-adult gerbils (blue shades), and the bottom row (**D**–**F**) shows data for quiet-aged gerbils (brown shades). Each column of graphs represents a different cochlear location: (**A**,**D**) 1 kHz, (**B**,**E**) 2 kHz, and (**C**,**F**) 16 kHz. Different symbols and color shading separately illustrate data for synapses located on the IHC’s modiolar half and its pillar half. There was a positive correlation between pre- and postsynaptic volumes for all subgroups. For descriptive purposes, the number of synapses analyzed and Spearman’s rho are indicated.

**Figure 7 ijms-25-02738-f007:**
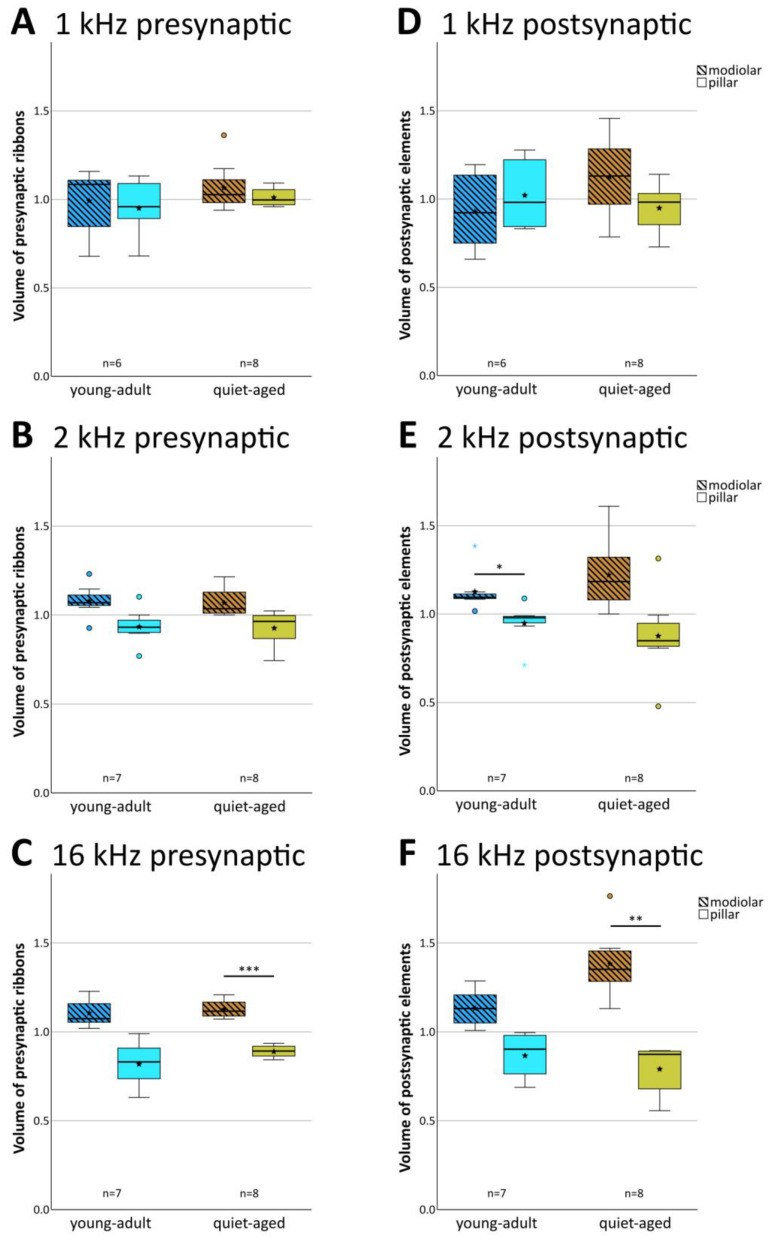
Normalized pre- and postsynaptic volumes were unaffected by aging. The left column of boxplots (**A**–**C**) shows data for presynaptic ribbon volumes, and the right column (**D**–**F**) shows data for volumes of postsynaptic GluR2 patches. Each panel compares the normalized volumes of modiolar-located (hatched boxes) and pillar-located (clear boxes) synaptic elements for young adults (blue shades) and quiet-aged gerbils (brown shades). Boxes indicate the median ± interquartile range, whiskers 1.5 times the interquartile range, outliers beyond those are shown as circles (>1.5, ≤3 times interquartile range) and colored asterisks (>3 times interquartile range). Stars indicate means. Black asterisks above the boxes indicate statistically significant differences between data for the modiolar half and the pillar half of the IHCs, as indicated by line brackets (* *p* ≤ 0.05, ** *p* ≤ 0.01, *** *p* ≤ 0.001).

## Data Availability

The data presented in this study are available on request from the corresponding author. The data are not publicly available due to insufficient resources for proper data annotation.

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
