# Peer review of "Cochlear Ribbon Synapses in Aged Gerbils"

_ijms, 2024, doi:10.3390/ijms25052738_

Round 1
Reviewer 1 Report
Comments and Suggestions for Authors
It is in the focus of hearing research how the inner hair cells (IHCs) transfer their sound-generated information to the primary auditory neurons, how this process is influenced by the age and what structural elements and changes underlie them. This is a nice follow-up study in this subject from the last/corresponding author’s lab to extend their results on age-dependent loss of the cochlear IHC-auditory fiber synapses [Ref 39] and on the gradients in the normalized volumes of synaptic elements along the modiolar-pillar axis of IHCs [Ref 15] in gerbil. In their submitted work, they confirmed the known age-dependent loss of synapses and showed the disproportionate loss of synapses on the modiolar side at higher frequency (16 kHz), which is consistent with previous observations in gerbil and strengthen the clarity of our view on this question; what is not the case with mice. They also showed that the larger volume of pre- and postsynaptic elements on the modiolar location, they proved previously, is more prominent toward the base of the cochlea. And this volume gradient is not affected by the age.
The manuscript is solid, but a more straightforward presentation of the data would improve the clarity and credibility of the manuscript (please, see the details in the Questions and comments).
Questions and comments:
1.) It is impossible or at least very-very hard for the reader to figure out which data came from where. A summary table collecting the information on used animals (sacrificed for this or the previous studies), their age, which of their cochlea / cochlear location provided a certain data would help a lot for the reader. A (seeming?) discrepancy like 13 animals in the “old” group (line 373-374) then “majority of cochleae from old gerbils (12 of 14)” (line 403-404) would also disappear, leaving the reader in no doubt as to whether 13 or 14 old animals were used. Moving the methods-like info from the Results (lines 102-111) to the Materials and Methods may also help.
2.) Please, define the axis (concerning Fig1/2 in general and the modiolar-pillar axis) and enhance understandability of the spatial orientations! An XYZ vector inset, like in Liberman, J Neurosci 2011, Fig.1 could be perfectly helpful. Giving the synapse positions on IHC in the 0 – 360° range/system, like at Merchan-Perez & Liberman, J Comp Neurol, 1996, could make much easier to understand the location of a synapse on an IHC than using the “relative position along IHC modiolar-pillar axis” concept. “as in panel A (left image) and rotated by 90° (right image, modiolar facing to the left as indicated)” - Why does 90° rotation of the modiolar plane results in the pillar plane/side? Why not 180°? Or “projection was rotated 107° around the x-axis” type of statements would be easier to understand for the reader.
3.) Fig1/2: On my downloaded version – both on printout and on the monitor – color coding is simply bad. Far from the nice one that can be seen e.g., in one of the authors previous paper [Ref 15]. With such a bad color coding, it is hard to believe that the identification of individual pre- or postsynaptic elements or their overlap is possible. Of course, this critique does not apply if the problem only concerns my downloaded version.
4.) Why do the authors show the medians on Fig3/5 when the Y axis label says “Mean” and for the statistical evaluation they also used the means? (“For the analysis of synapse numbers, mean values were used for each frequency location”) I would prefer to see the means. At least to see the means, as well. (And individual data points, as they appear on some figures.) This also applies to Fig7, where the Y axis label is not correct, in my opinion. Graphs show the ‘Volume of presynaptic ribbon/postsynaptic elements (normalized to the median volume)’, not the “Mean volume of presynaptic ribbon/ postsynaptic elements”.
5.) Please, show the number of experiments (n) on Figs (at least in the legends), as they are given in Fig. 6. Knowing the respective number of values in a certain tonotopic location and the number of respective cells would be very informative and useful, too.
6.) Fig4: Appearing several synapses on the IHC upper side in D and E (vs. A and B) and their disappearing on F (vs. C) is a noticeable difference. What could be the reason of this dispersion and contraction of synaptic contacts on IHCs and what could underlie this interesting tonotopic dependence?
7.) Fig6: Synaptic volumes normalized to the median volumes are plotted on the graphs and the normalized values cover two orders of magnitude (0.1 – 10). Does this mean that absolute volumes of pre- and postsynaptic elements are distributed in such a large scale?
Minor points and suggestions:
1.) Using “old” and “quiet-aged” as synonyms is OK in the text, but may not be the best solution to mix them on the figures.
2.) Line 21-22: Delete one of the “about 20%”!
3.) Line 354: Correct the typing error of “n” to in!
Reviewer 2 Report
Comments and Suggestions for Authors
This study attempts to determine specificity in changes in auditory nerve fiber/inner hair cell (IHC) ribbon synapses that occur with age in Mongolian gerbils, modeling age-related cochlear degeneration and loss of hearing. The experimental approach is comprised of tissue immunohistochemical labeling followed by fluorescence microscopy and image analysis with a Matlab algorithm previously written by this group. The primary finding is that age-related synapse loss in the gerbil cochlea, despite occurring in all locations studied, was significantly greater for the modiolar region of the IHC’s in the basal cochlea. This region is associated with 16 kHz frequencies, suggesting a disproportionate effect of aging on the loss of low-spontaneous rate synaptic innervation at high frequencies.
Unfortunately, this conclusion is based on a severely flawed experimental design, with two critical issues. The first major issue is the use of older, previously analyzed synaptic immunohistochemistry data. From the authors’ own admission: “The data for young-adult gerbils previously published by Zhang et al. [15] were incorporated here and provided about half of the young-adult sample (75 of 164 IHC). From the sample of old gerbils, synapse counts from 17 of the 24 cochlear locations analyzed here were published previously [39]. However, since the present study aimed to add data on the volumes of synaptic elements in aged IHC, the analysis was carried out de novo and included more hair cells from each specimen.” Antibody-based techniques such as immunohistochemistry are infamously sensitive, especially for quantitative analyses. Samples that are harvested and stained in different batches can have widely varying baselines, necessitating either the batch processing and imaging of all samples within a very short period of time (i.e. weeks-months) or the use of normalization techniques that can have undesirable consequences on statistical comparisons. Neither of which seems to have been the case with this study as reported. Furthermore, it is unclear whether the additional de novo analysis that was done for the older gerbil samples required new imaging. If so, these images would presumably have been taken from samples that were already several years old (the referenced citation is from 2021) and would have undergone a significant loss of fluorescent signal during that time. Even if the de novo analysis was performed with images that were taken around the time of the initial tissue processing, they would have been added to a pool of images that were taken from newly harvested and processed tissue, resulting in an incongruous comparison.
The second major issue is that the young and old gerbil tissue samples were not processed the same way, yet they are being compared statistically in this study. The Methods state that the old gerbil samples were treated with an autofluorescence quencher that was not used in processing the young samples. This procedural difference would not be a big issue if the young and old groups weren’t being directly compared, but that is precisely the focus of this study. As such, these comparisons were not made between equivalent groups, and the actual mechanisms underlying any found differences would therefore be indeterminable.
If the authors are intent on keeping the previously published young gerbil data in this study, they must conduct additional analyses between that data and the newly acquired data, showing that there are no significant differences between groups when other factors (e.g. cochlear location, modiolar-pillar axis position) are taken into account. Secondly, they cannot include the statistical comparisons between young and old gerbils due to the difference in methodology. Each age group would have to be presented as an independent study/result. Though this potentially weakens the findings of the overall study, it will at least alleviate the issue that the current experimental design does not justify the comparisons as currently presented.
Round 2
Reviewer 2 Report
Comments and Suggestions for Authors
The authors have provided additional clarifications and comparisons that have alleviated my major concerns from the previous version of the manuscript. I have no further issues.